# Improved Charge Transfer Contribution by Cosputtering Ag and ZnO

**DOI:** 10.3390/nano10081455

**Published:** 2020-07-25

**Authors:** Bingbing Han, Shuang Guo, Sila Jin, Eungyeong Park, Xiangxin Xue, Lei Chen, Young Mee Jung

**Affiliations:** 1Key Laboratory of Preparation and Applications of Environmental Friendly Materials, Jilin Normal University, Ministry of Education, Changchun 130103, China; hanbingbing678@163.com (B.H.); xuexiangxin0804@163.com (X.X.); 2Department of Chemistry, Institute for Molecular Science and Fusion Technology, Kangwon National University, Chunchon 24341, Korea; guoshuang@kangwon.ac.kr (S.G.); jsira@kangwon.ac.kr (S.J.); egpark@kangwon.ac.kr (E.P.)

**Keywords:** SERS, SPR, CT, EM, Ag/ZnO film

## Abstract

A two-dimensional polystyrene microsphere array cosputtered with Ag and ZnO was designed for evaluating surface-enhanced Raman scattering (SERS) activity. The surface plasmon resonance (SPR) and SERS properties were significantly changed by the introduction of ZnO into the Ag film. By increasing the Ag sputtering power, a redshift of the SPR peak was obtained. Moreover, improved SERS activity occurred because of the electromagnetic (EM) contribution from the increasing Ag content and the charge transfer (CT) contribution from the introduction of ZnO. More importantly, the Hall effect was employed to evaluate the carrier density effect on the SERS contribution of the Ag/ZnO film. The increase in the carrier density as the Ag sputtering power increased indicated an increasing number of free electrons stored in the Ag/ZnO film, which was accompanied by improved EM and CT contributions.

## 1. Introduction

With the development of nanoscience and laser technology, numerous development opportunities exist for surface-enhanced Raman scattering (SERS) in interface and surface investigations and life science applications [1,2,3,4]. SERS is a surface or interfacial phenomenon that is sensitive to variations in the environment of the substrate surface. When a laser irradiates a metal and a dielectric layer, the surface plasmons in the metal collectively oscillate with unique surface plasmon resonance (SPR) characteristics, thereby resulting in higher SERS activity. Thus, electromagnetic (EM) enhancement is induced [5,6,7]. Generally, the SPR is affected by the excitation of thermal electrons, thermal effects and optical near-field enhancement [8]. Generally, Au, Ag, Cu, and a few transition metals are widely used as SERS-active substrates that provide higher SPR with laser excitation [9,10]. The greatest advantage is that SERS technology can reach the level of single molecule detection by using metal nanoparticles (NPs) [11,12]. However, the SERS effect is also faced with two constraints [13,14]. First, the development of a new SERS substrate that simultaneously has the advantages of stability, a good reinforcement effect, and high reproducibility is needed. Second, finding a suitable method to conduct in-depth research on the characteristics and essence of the two SERS enhancement mechanisms is necessary. Therefore, multimaterial composites, such as metal/semiconductor composites or inorganic hybrid nanostructures, were developed as SERS-active substrates that exhibited excellent SERS properties.

Semiconductor materials provide many opportunities for SERS because of their inherent physical and chemical properties [15,16,17,18]. Therefore, the preparation of metal-semiconductor SERS-active substrates with high sensitivity and controllability has attracted great attention. A strong electric field forms near the nanocomposite or hybrid nanostructure surface under SPR excitation. In addition, the plasma-induced charge separation process occurs through the traditional thermal electron transfer mechanism between the metal and semiconductor [19]. Meanwhile, a new plasma attenuation path is formed at the metal/semiconductor interface, which generates electron/hole pairs at the metal/semiconductor interface. Therefore, the electrons transit back to the metal, providing a possible charge transfer (CT) process [20,21].

ZnO is a semiconductor that has a specific electron transfer capability and good adsorption characteristics; furthermore, ZnO exhibits a wide band gap [22]. Optimized ZnO doping can reduce the oxidation and improve the stability of Ag, avoid the generation of background fluorescence and improve the quality of SERS. To date, the SERS enhancement mechanism for the interaction between semiconductors and metals is still not completely clear. Therefore, the design and manufacture of Ag-ZnO composite materials as SERS-reinforced substrates with simple and controlled metal SPR is very important. Herein, we cosputtered Ag and ZnO with different doping ratios on a polystyrene (PS) template, which then exhibits tunable SPR properties. SERS was employed to evaluate the effects of the ZnO introduced into the metal-semiconductor system. The SERS activity can be easily controlled to compare the different behaviors that occur for different doping ratios of Ag and ZnO under 514 and 633 nm illumination. In addition, the CT process on Ag/ZnO-4-mercaptobenzoic acid (MBA) was studied to evaluate the contributions of the as-prepared material to SERS enhancement.

## 2. Experimental Section 

### 2.1. Chemicals

Ag and ZnO targets (99.99%) were obtained from Beijing TIANRY Science and Technology Development Center (Beijing, China). PS microspheres (200 nm, 10% *w*/*w*) were purchased from Bangs Laboratories Inc. (Fishers, IN, USA). MBA and absolute ethanol were purchased from Sigma-Aldrich Co., Ltd. (Shanghai, China) and Beijing Chemical Work (Beijing, China), respectively. All chemicals were used without further purification.

### 2.2. Preparation of the Ag/ZnO 2D Ordered Array

Two-dimensional (2D) ordered arrays were fabricated by the self-assembly method. The preparation process of Ag/ZnO-coated 2D arrays is shown in Scheme 1. Briefly, a uniform array was formed on the surface of water by diluting 200 nm PS microspheres in a volume ratio of 1:1 with absolute ethanol, and then, the solution was added to the water surface at a constant rate. Next, a hydrophilic silicon wafer (size: 1 × 2 cm^2^) was used to remove the array. Subsequently, Ag and ZnO were coated on the prepared PS arrays by employing a magnetron sputtering system (ATC 1800-F, AJA International Inc., Scituate, MA, USA). In the cosputtering process, the Ag target and ZnO target were used as the magnetic target and nonmagnetic target, respectively, the angle between the substrate and the target was 10°, and the distance between the substrate and the target was 20 cm. The sputtering power for Ag was 10, 20, 30, and 40 W, which for ZnO was 100 W, and the cosputtering time was 5 min under Ar gas conditions. The Ar gas flow rate was 9 sccm (standard cubic centimeters per minute), and the working pressure was 5.8 × 10^−3^ mTorr. The working pressure and the base pressure were 6 × 10^−1^ Pa and 2 × 10^−5^ Pa, respectively. MBA, which was selected as a probe molecule, was dissolved in absolute ethanol at a concentration of 10^−3^ M, and the Ag/ZnO-coated PS arrays were immersed in the MBA solution for 1 h to obtain SERS spectra.

### 2.3. Characterization of the Ordered Arrays

The Ag/ZnO-coated PS arrays were observed by scanning electron microscopy (SEM, JEOL 6500F, accelerating voltage of 200 kV, Tokyo, Japan). A Shimadzu UV−3600 spectrometer (Toyko, Japan) was used to obtain ultraviolet-visible (UV-Vis) absorption spectra. The carrier density of the Ag/ZnO film was tested by a Hall effect detector (Lakeshore, 775 HMS Matrix, Columbus, OH, USA). Raman spectra were acquired by a confocal Renishaw Raman System 2000 microscope spectrometer (irradiation at 514/633 nm, London, UK). Under the excitation wavelengths of 514 and 633 nm, the excitation power was 20 and 17 mW, the attenuation power was 50%, the acquisition time interval was 10 s and the spot diameter was approximately 1 μm.

## 3. Results and Discussion

The morphology of the substrate is an important parameter that affects the properties. Herein, a 2D ordered PS microsphere particle array was fabricated by self-assembly technology, and Ag/ZnO 2D ordered arrays were prepared by the double-target cosputtering method. Briefly, Ag and ZnO were sputtered for 5 min, and for samples with different compositions, the Ag sputtering power was 10, 20, 30, or 40 W, while the ZnO sputtering power was 100 W. The cosputtering process of the Ag/ZnO ordered arrays is shown in Scheme 1. The morphologies of Ag/ZnO coated on the PS arrays under different Ag sputtering powers are shown in Figure 1. Figure 1e presents an SEM image of a bare PS array. The diameter of the PS microspheres is approximately 200 nm, and the microspheres are uniformly stacked. Figure 1a–d show that as the Ag sputtering power increased, the PS surface roughness gradually increased, which is caused by the growth and nucleation of Ag during the sputter deposition process [23]. Gaps between the microspheres were gradually filled, which will strongly affect the position and intensity of “hot spots” in the ordered array [24]. In addition, the elemental distribution of Ag, Zn, and O covering PS was confirmed by SEM elemental mapping, indicating that the relative content of each element in the composite film material changed with increasing Ag sputtering power. Figure 1f–i shows cross-sectional images corresponding to Figure 1a–d, respectively. The array thicknesses were observed to be 218, 230, 244, and 260 nm, which were basically consistent with the profilometer test results. Changes in the thickness of the composite will affect its optical and electrical properties.

As shown in Figure 2, UV-Vis absorbance spectra were obtained to explore the optical characteristics of the Ag/ZnO ordered arrays. These substrates show strong absorbance in the visible region. With increasing Ag content, the position of the band assigned to PS, which is marked with crosses, almost did not change, but the intensity was weakened, indicating that the thickness of the coating on the surface of the PS microspheres was increasing and that the coating gradually covered the PS microspheres. The absorption peak marked with circles at 260 nm could be attributed to the SPR of ZnO, which shows a weakening trend with increasing Ag content. The band at 330 nm, which is marked with diamonds, belongs to the dipole vibration peak of Ag [25]. The band marked with stars belongs to the SPR of the Ag/ZnO array. As the Ag content increased, the charge of the Ag/ZnO array was redistributed, which causes the SPR peak to redshift and show a broadening trend. When the Ag content was sufficiently high, this peak was close to the absorption peak of the pure Ag film. In addition, the band at approximately 400–700 nm indicates the SPR between the dipole and the Ag/ZnO array. Notably, the system had a wide absorption band in the 400–700 nm region, which gradually redshifted and broadened with increasing Ag content, approaching the absorption band of the pure Ag film (>700 nm). The observed redshift and broadening of the absorption bands can be attributed to interparticle coupling and the changed electronic environment around the Ag/ZnO array. As the Ag content in the Ag/ZnO film increased, Ag and ZnO accumulated on the surface, which induced in-plane coupling that redshifted the SPR peak.

To investigate the SERS characteristics of the Ag/ZnO films, MBA with a concentration of 10^−3^ M was selected as the probe molecule to explore the SERS spectra when adsorbed on the substrate surface. SERS spectra of MBA adsorbed on the Ag/ZnO films under 514 and 633 nm laser excitations are shown in Figure 3a,b, respectively. SERS bands of MBA were observed at approximately 997, 1018, 1073, 1180, 1363, and 1582 cm^−1^, and the assignments of these bands are listed in Table 1 [26]. Pure ZnO films had little enhancement effect on MBA. However, due to the high SPR of the Ag material in the visible region, it can make a great contribution to the EM field and to a high SERS activity. [27]. Therefore, Figure 3a,b show that as the Ag sputtering power increases, the SERS intensity is significantly enhanced. At the same time, the intensity of the b_2_ (nontotally symmetric) vibration modes was also significantly enhanced. More importantly, compared with the same amount of pure Ag film, the intensity of the b_2_ vibration modes of the Ag/ZnO composite was stronger. According to the CT model proposed by Lombardi et al., only the totally symmetric vibrational modes of the probe molecules can be enhanced by the contribution corresponding to the Franck–Condon principle, while both totally and nontotally symmetric vibrational modes of the molecules are expected to be enhanced via the Herzberg–Teller effect [28]. That is, the intensity of the b_2_ vibration modes of MBA adsorbed on the surface of Ag/ZnO is strongly enhanced by the CT mechanism based on the Herzberg–Teller contribution. Therefore, we suggest that the SERS enhancement is based on the synergistic effect of the EM and CT contributions.

The degree of CT (ρ_CT_) was employed to quantitatively estimate the effect of the CT resonance contribution on the overall SERS intensity, which was proposed by Lombardi et al. [28]. The value of ρ_CT_ (*k*) is determined according to the following equation:(1)$$\rho_{\mathrm{CT}}\left(k\right)=\frac{I^{k}\left(\mathrm{CT}\right)-I^{k}\left(\mathrm{SPR}\right)}{I^{k}\left(\mathrm{CT}\right)+I^{0}\left(\mathrm{SPR}\right)}$$
where *k* is an index for identifying individual Raman bands. *I^k^*(CT) is the intensity of Raman bands affected by the CT resonance. The band at 997 cm^−1^ (b_2_ mode) was nontotally symmetric (the intensity is denoted by *I^k^*(CT), which was enhanced by CT resonance, providing an additional SERS enhancement, excluding the SPR contribution. *I^k^*(SPR) was usually small or equal to zero. The totally symmetric band at 1180 cm^−1^ (a_1_ mode) was mainly contributed by SPR (the intensity is denoted by *I^k^*(SPR); in this case, *I^k^*(SPR) = *I*^0^(SPR). Figure 4a shows the calculated ρ_CT_ under 514 and 633 nm laser excitations. As shown, with increasing Ag sputtering power, ρ_CT_ gradually increased; subsequently, when the Ag sputtering power was 40 W, ρ_CT_ ˃ 0.5, thus indicating that the CT resonance contribution was significantly increased under 514 nm laser excitation. However, under 633 nm excitation, ρ_CT_ did not increase significantly when the Ag sputtering power was 10–30 W, while when the Ag sputtering power was 40 W, ρ_CT_ shows a significant increase. To further analyze this phenomenon, we explored the CT mechanism of the Ag/ZnO/MBA systems.

The energy level schematic of the Ag/ZnO/MBA systems for the CT mechanism is shown in Figure 4b. The Fermi level of Ag was located at −4.26 eV with reference to the vacuum level [29]. The highest occupied molecular orbital (HOMO) and the lowest unoccupied molecular orbital (LUMO) levels of MBA were situated at −6.24 and −1.68 eV, respectively [30]. The conduction band (CB) and valence band (VB) of ZnO were situated at −3.80 and −7.17 eV, respectively. [29] The energy barrier from the Fermi level of Ag to the LUMO of MBA was 2.58 eV, which was higher than the incident photon energy (514 nm, 2.41 eV, and 633 nm, 1.96 eV). Thus, the incident photons did not have sufficient energy to directly transfer the excited electrons of Ag into the LUMO of MBA. In this case, the SERS enhancement effect from the CT contribution could not be observed. In the Ag/ZnO/MBA system, ZnO is equivalent to a “bridge” that indirectly transfers the electrons in Ag to the LUMO level of MBA. In this case, the energy barrier from the Fermi level of Ag to the CB of ZnO was 0.46 eV, and that from the CB of ZnO to the LUMO of MBA was 2.12 eV; thus, the incident photons (514 nm, 2.41 eV) provided sufficient energy to indirectly excite electrons from Ag through the CB of ZnO to the LUMO of MBA. Therefore, under 514 nm laser excitation (2.41 eV), CT could occur, and the selective enhancement of CT could be observed in the SERS spectrum of the Ag/ZnO/MBA system. However, under 633 nm laser excitation, the excitation light energy (1.96 eV) was insufficient to excite the CB electrons of ZnO to the LUMO level of MBA, but a significant CT enhancement was observed when the Ag sputtering power was 40 W. We suggest that this phenomenon was attributed to a plasma-induced CT enhancement. That is, when the Ag content was sufficient, the carrier density at the interface of the Ag/ZnO system was sufficiently large, and the excitation provided by the light will produce a strong plasmon resonance phenomenon; this phenomenon promoted the transition of electrons to a higher energy level. Therefore, we could confirm that the SERS intensity is attributed to the synergistic effect of the EM and CT contributions.

To further analyze the effect of carrier variation on SERS enhancement, we employed the Hall effect, which can be used to detect the carrier density and mobility of a metal or semiconductor [31]. The Hall effect can be quantitatively analyzed to determine the effect of the doped ZnO content in the Ag film on the carrier density. The carrier density in the Ag/ZnO system can be controlled by varying the ZnO content, and the results of the Hall effect are displayed in Table 2. The carrier density in the Ag/ZnO film increased as the Ag sputtering power increased, which indicates that an increased number of free electrons were stored in the Ag/ZnO film. Therefore, CT was possible because of the variation in the carrier density. As shown in Figure 5, the relationship between the CT and the carrier density indicates that with increasing carrier density, the CT synchronously increased. According to Figure 3 and Figure 4, with increasing Ag sputtering power, the SERS intensity of the a_1_ and b_2_ modes increased significantly. This result means that the higher the carrier density is, the more free electrons that are involved in plasma resonance and CT enhancement.

## 4. Conclusions

We employed the cosputtering technique to fabricate an Ag/ZnO film and discussed the spectroscopic performance of Ag/ZnO films with various amounts of Ag. The observed SERS enhancement of MBA on the Ag/ZnO films was due to the increase in free electrons, which was caused by the increase in Ag content. The carrier density in the Ag/ZnO film was utilized as an effective measure to verify the increase in the SERS enhancement. The increased carrier density induced high EM and CT contributions from the Ag/ZnO film to the MBA. Thus, the results show that the carrier density is an important factor in increasing the SERS enhancement. This study provides a new idea for exploring SERS-active substrates for use in photocatalysis and optoelectronic device applications.

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
