# Peer review of "Improved Charge Transfer Contribution by Cosputtering Ag and ZnO"

_nanomaterials, 2020, doi:10.3390/nano10081455_

Round 1
Reviewer 1 Report
The paper presents a study on a hybrid ZnO/Ag substrate for SERS applications. The research presented is certainly interesting but I cannot support the pubblication of the paper in the present from. Many aspects of the paper should be improved:The introduction is very confused and not well organized. Extensive rephrasing of english language should be done to make it readable.
The part related to the discussion of the visible spectra should be revised and the conclusions supported by some arguments. For example the SPR band (blu circle, black Line spectrum) for ZnO sputtering only is hard to believe. The most important band in SERS activation is the plasmon band occurring below 500 nm, and should be discussed further.
The SERS enhancement reported in fig.3 seems a nice results, but It world be meaningful if the authors report the quantity/concentration of the probe molecule MBA, and a comparison with a spectrum without Ag sputtering.
Without extensive revision of the introduction, visible spectra discussions, and the complete presentation of the SERS results, I can not recommend pubblication of the present paper.
Minor point: In figure 1, the authors should mention to which substrate they are referring in panel e,f,g.
Author Response
We appreciate the Reviewer’s comments on this manuscript, which helped us significantly revise the manuscript.
According to the comments of Reviewer, we prepared a highlighted revised version documenting all changes made. Our point-by-point response to the Reviewer's comments is attached herein.

Reviewer 2 Report
The manuscript in good scientific manner show impact of the charge transfer on the SERS enhancing properties. Based onto the SERS spectra of chosen analyte Authors introduce the Franck-Condon principle, as well as Herzberg-Taller effect. I recommend this article to be published.
Two remarks:
- I advise to move the Table S1 and S2 from Supplementary Materials, to the main text of the manuscript.
- For the Figure 5, some additional explanation (meaning of the constructed line between points) are necessary e.g. “guide for an eye”.
Author Response

(The authors gave the same response as above.)

Reviewer 3 Report
In this contribution Han et al. produced surface-enhanced Raman scattering (SERS) substrates by the co-sputtering of Ag and ZnO on a two-dimensional polystyrene microsphere array. In my opinion all experiments have been carried out correctly and their interpretation is also correct. Authors suggest that the surface plasmon resonance (SPR) and SERS properties were significantly changed by the introduction of ZnO into the Ag film. I am pretty sure that it is true. Unfortunately, the authors have not shown results obtained for the films produced from pure silver (without ZnO). They always show results obtained for Ag/ZnO and pure ZnO films. In my opinion comparison of the results obtained for pure Ag and for Ag/ZnO films would significantly improve the quality of this work. Therefore, I suggest its revision.
Author Response

(The authors gave the same response as above.)

Reviewer 4 Report
The work by Han et al. (Improved Charge Transfer Contribution by Cosputtering Ag and ZnO) reports cosputtering of Ag and ZnO onto the two-dimensional polystyrene microsphere array and their employment as surface-enhanced Raman scattering (SERS) platform. The authors investigated the the effect of Ag sputterring power on surface plasmon resonance (SPR) and SERS properties of the substrate. It was detected that the ÅŸncrease of the Ag content led to increase in carrier density and resultant charge transfer contributions.
In general, the manuscript is well-written and easy-to-follow. I think the report provides valuable information to this researh field. I recommmend the publication of the manuscript in the Nanomaterials, after some major points given below are addressed.
Line 53-54
Please fix this statement.
Please define the sputtering gas in the Experimental Section. What is the deposition rate for the Ag and ZnO? Is there any angle betwen the targets and subtrate? What is the concentration of PS microspheres? Please provide detailed information in this section.
Characterization section does not provide adaquate information. Please define Raman laser power, spot size, acquisition time etc.
What is the silver sputterring power for the substrate in Figure 1e-g? For the comparison, please add the SEM images of the bare PS arrays. Also, EDS mapping for the all silver sputterring powers would be informative to make certain conclusions. I wonder if the author could provide the elemental content (in mass %) of the thin film. These data would be highly informative to explain Uv-vis data given in Figure 2 and SERS activities. I think the authors can determine the thickness of the composite film by using untreated (flat) wafer under the same sputtering conditions. Cross-section imaging via SEM or profilometer would be helpful.
Line 178 and 183
Please give the corresponding wavelengths.
Author Response

(The authors gave the same response as above.)

Round 2
Reviewer 1 Report
none
Reviewer 4 Report
The authors did great development in the revised version of the manuscript. I recommend the publication of the report in its present form.